# Current Agronomic Practices, Harvest & Post-Harvest Processing of Soybeans (*Glycine max*)—A Review †

Ondulla T. Toomer [1,*], Edgar O. Oviedo [2], Muhammad Ali [2], Danny Patino [2], Michael Joseph [2], Mike Frinsko [3], Thien Vu [1], Pramir Maharjan [4], Ben Fallen [5] and Rouf Mian [5]

1. Food Science & Market Quality and Handling Research Unit, Agricultural Research Service, United States Department of Agriculture, Raleigh, NC 27695, USA
2. Prestage Department of Poultry Science, North Carolina State University, Raleigh, NC 27695, USA
3. Marine Aquaculture and Research Center, North Carolina State University, N.C. Cooperative Extension, Jones County Center, Trenton, NC 28585, USA
4. Department of Agriculture and Environmental Sciences, Tennessee State University, Nashville, TN 37209, USA
5. Soybean and Nitrogen Fixation Research Unit, Agricultural Research Service, United States Department of Agriculture, Raleigh, NC 27695, USA
* Correspondence: ondulla.toomer@usda.gov
† Mention of a trademark or proprietary product does not constitute a guarantee or warranty of the product by the U.S. Department of Agriculture or North Carolina Agricultural Research Service, nor does it imply approval to the exclusion of other products that may be suitable. USDA is an equal opportunity provider and employer.

**Abstract:** Globally, soybeans are grown to meet the needs for animal and human nutrition, oil extraction, and use in multiple industrial applications. Decades of soybean research, innovative farming methods, and the use of higher yielding resistant seed varieties have led to increased crop yields. Globally, soybean producers have utilized enhanced processing methods to produce nutritious high-quality meal and extracted oil for use in animal feed and within the food industry. Soybeans contain highly digestible proteins and are processed using various mechanical and chemical techniques to produce high quality animal feed ingredients. Defatted soybean meal (DSM) is usually prepared by the solvent extraction process of soybeans, whereby almost all oil content is removed. When oil is not extracted, full-fat soybean meal (FFSBM) is created. This form provides an excellent source of dietary energy by retaining the lipid component and is very useful in animal feeds by reducing the need for adding exogenous lipids. However, some anti-nutritional factors (ANF) are present in FFSBM if not properly heat treated before inclusion in the finished feed. These ANF adversely affect the internal organ function and overall growth performance of the animal. Among these ANF, protease inhibitors are most important, but can be readily destroyed with optimal thermal processing. However, if the process protocols are not followed precisely, excessive heat treatment may occur, resulting in both reduced protein quality and amino acid bioavailability in the meal. Conversely, insufficient heat treatment may result in the retention of some ANF in the meal. Thermally resistant ANF can be greatly reduced in the bean and meal when dietary enzyme supplementation is included in the finished feed. This approach is cost-effective and most commonly utilized commercially. After processing, the soybean meal quality is often measured using in vitro methods performed at commercial analytical laboratories to assess the nitrogen solubility index (NSI), protein dispersibility index (PDI), urease activity (UA), and protein solubility in potassium hydroxide. Once properly processed, FFSBM or DSM can be utilized optimally in the diets of poultry and aquaculture to enhance the economic viability, animal nutrition, production performance, and the quality and nutritional value of the meat and/or eggs produced.

**Keywords:** precision farming techniques; soybean meal quality; extrusion processing; high-oleic soybeans





## 1. Introduction and Agronomic Practices

Soybeans are a leguminous crop, primarily grown for oil extraction, leaving the remaining soy "cake" as a source of highly digestible amino acids [1]. Interestingly, only 2% of the globally produced soybean meal is used for food, with the remaining 98% utilized for the nutrition of animals [2], with poultry consuming 64% of the U.S.-produced soybean meal, followed by swine at 24%, beef and dairy cattle at 10%, and other animals, such as aquaculture and companion animals, at 2% [3]. Soybeans are primarily grown and harvested in the northern Midwest of the U.S., with Illinois, Iowa, and Minnesota being the top soybean producers in the U.S. [4]. In more temperate areas of the Midwest, soybeans can be double cropped with winter wheat, whereby both crops can be consecutively grown and harvested on the same land within the same year [5].

A great deal of preparation is required prior to planting soybeans in the field. First, soil fertility must be determined. This is conducted by collecting and analyzing soil samples to ensure sufficient nutrients are available for the crop to thrive, resulting in a maximized crop yield, while minimizing any negative environmental impact. Generally, the primary field macronutrient requirements for soybeans are the following: nitrogen, phosphorous, and potassium. Secondary, but still critical, micronutrients include sulfur, calcium, magnesium, zinc, manganese, boron, iron, and copper [6]. The soybean fertilization is usually unnecessary due to the nitrogen fixation capabilities of the symbiotic bacterium Bradyrhizobium japonicum. This common soil bacteria naturally performs a process of nitrogen fixation, resulting in excess nitrogen to be available to support good plant growth [7]. Significantly, 25–75% of the nitrogen in mature soybeans has been shown to originate from symbiotic nitrogen fixation [8]. Additionally, it has been well-documented that soybeans prefer a slightly acidic soil, with a pH ranging from 6 to 7, which enhances the nutrient bioavailability to plants. Commonly, lime is added to the soil to increase the pH, while the application of elemental sulfur is applied to lower the pH [6].

Critical components of field management also include proper row spacing and seed density for newly planted soybeans to ensure the maximum yield. Typically, narrow rows and high plant densities correspond with accelerated canopy closure, which suppresses weeds [9]. Irrigation planning is also a vital management tool to achieve the optimal soybean yield. Traditionally, an irrigation schedule is developed based upon historical weather records, predicted weather forecasts, or a combination of these factors [6]. However, current soybean producers improve crop yields with the use of precision farming techniques and advanced information technology, which can detect inter- and intra-field variability. As an example, enhanced sensors can now accurately collect the real-time moisture and temperature data of the soil and environment in the field [10]. In addition, satellite imaging and robotic drones provide soybean producers with real-time imaging of individual plants and conditions [10].

A major challenge of efficient soybean crop production involves pest and weed management. Up to 80% of the annual soybean crop damage is due to numerous entomological pests, including two-spotted spider mites, aphids, stinkbugs, loopers, beetles, and the kudzu bug. The degree and severity of the resulting losses vary from year to year and by geography [11]. Traditionally, insecticides have constituted a large portion of the pest management arsenal, as the primary method of control. In addition, the interference of unwanted and destructive weeds causes several million U.S. dollars of economic losses each year for soybean growers [12]. Weeds, such as the common water hemp (Amaranthus rudis), Canadian horseweed (Conyza canadensis), giant ragweed (Ambrosia trifida), ivy-leaf morning glory (Ipomea hederacea), common cocklebur (Xanthium strumarium), Johnsongrass (Sorghum halepense), and pigweed (Amaranthus spp.) compete for available nutrients, field space, and other vital resources [12]. Today, integrated pest management (IPM) is the most common strategy used to manage insect and weed problems. IPM focuses on the use of herbicides in conjunction with pragmatic production practices and the targeted use of herbicide resistant soybean cultivars, such as glyphosate resistant Roundup Ready soybean varieties for successful cultivation [12]. "Roundup" is a common weed

killer containing glyphosate that prevents plants from making the proteins they need to survive; therefore, Roundup is considered a "broad spectrum" herbicide [13]. Roundup not only kills weeds but may also damages the crops of interest. Hence, in 1996, Monsanto genetically engineered the "Roundup Ready Soybeans", which are resistant to glyphosate [14]. Field weed management practices with pre- and post-emergent herbicide applications have been shown to prevent damage to soybean plants, while preventing weeds [9].

In addition to insects and weeds, soybeans are susceptible to a variety of fungal and bacterial diseases. Fungal and bacterial pathogens may reduce soybean crop yields up to 50% and 15–60%, respectively [11]. Globally, soybean crops are affected by five of the most prevalent plant pathogens. These disease-causing organisms vary by year and geographical location, and include H. glycines, Phytophthora sojae, Colletotrichum truncatum, Septoria glycines, and Phakopsora pachyrhizi [11]. H. glycines, commonly known as Soybean Cyst Nematode (SCN), causes more economic losses than any other soybean disease globally, resulting in up to 90% yield losses in some geographical regions [11].

The early detection of SCN is a key to managing and preventing the spread of this disease. However, the early detection of SCN is challenging because soybeans do not show any above-ground symptoms, unless there is significant observable damage. Consequently, there has been a great deal of research to identify SCN-resistant genes. In the last few years, some success has occurred, whereby the Rhg1 gene, an amino acid transporter, reduces the reproduction of the SCN and improves yield in soybean plants in fields that are infected with SCN [15]. Additionally, other research has identified molecular methods using DNA-specific quantitative real-time primer sets to detect SCN in artificially and naturally infected soil samples from soybean fields [16].

Within the last few decades, soybean crops within the northern central region of the U.S. were vulnerable to the "white mold" disease. This fungal pathogen originates in the soil and occurs as a white cottony, moldy growth produced from the fungus, Sclerotinia sclerotiorum [17]. White mold thrives in cool and moist conditions and weakens the plant as it spreads, reducing crop production, and/or causing plant death. Interestingly, Ohio soybean growers have had localized outbreaks of "white mold" every year since 2009 [18]. In these cases, diseased soybean plants appear wilted with grayish green leaves that later turn brown. The diseased plants that are present in high moisture conditions have stems with bleached white lesions covered with white fluffy mycelia and wilted grayish green leaves that later turn brown [18]. While no single field management tool is effective for the prevention and/or control of "white mold", this disease can be best mediated through the use of multiple field management tools, such as the use of "white mold" resistant soybean cultivars, the use of corn and wheat crop rotation to prevent pathogenic soil inoculum build-up over time, good weed management (several weeds are host to the pathogen), the use of Boscalid fungicide, and other production controls to limit the introduction of "white mold" to the field with infected equipment and/or seed [18].

Chemical pesticides have been shown to be very effective as a treatment strategy against all major soybean fungal and bacterial pathogens [6,11]. However, with growing consumer concerns regarding the use and impact on human health and the environment, research based innovative methods have been developed to effectively control many plant pathogens, pest, and weeds, without adverse environmental or human impacts. Moreover, these methods of IPM are successful as they are based on the specific and detailed knowledge of the disease and/or the pest life cycle, and its interaction within the environment. Hence, IPM reduces the use and environmental impact of pesticides, while also reducing the human direct and indirect exposure to chemical pesticides [6]. Production practices utilized in IPM may include altering planting dates, modifications in row spacing, using no-till fields, and the use of resistant soybean cultivars [11].

## 2. Soybean Harvest and Post-Harvest

When the soybean seeds mature, the pods and stem turn yellow in color and contain approximately 45–55% moisture [19]. Within four to nine days following the yellowing onset of maturity, the soybean pods turn brown in color and the moisture content will have reduced to about 33% [19]. Once the soybean pods finally turn brown in color, the soybeans are ready for harvest within four to five days, given favorable drying weather. Recent studies have shown it more profitable for soybean growers to harvest soybeans at 15% and 16% moisture than the prior practice to harvest them at 11–13% moisture. Harvesting at 14% is more profitable than harvesting at 12% [20]. Typically, the moisture levels are closely monitored during the harvest to prevent losses during the process of "combining", in which shattering losses are very high when moisture levels are drier, and the beans drop below 13% moisture [19]. In non-favorable drying conditions (wet and colder conditions), chemical desiccant drying sprays can be utilized to reduce the moisture in the pods, while simultaneously killing weeds [19]. The most-used soybean pre-harvest desiccants are paraquat, saflufenacil, and sodium chlorate [21].

Post-harvest, whole soybeans must be kept cool and dry and free of foreign debris. The soybeans' moisture fluctuates readily, with rapid gains and losses of moisture content. To manage these changes, storage bins must be well-aerated to control moisture and condensation with the use of drying or aeration fans [19]. Post-harvest soybeans are dried naturally in the field or artificially with no or low heat to a "safe low level of moisture (13%)" to ensure the preservation and quality of the beans during the long-term storage [22]. Low moisture levels are critical during the pre-harvest and post-harvest process to minimize the growth of aflatoxin-producing mold contamination: Aspergillus glaucus, Asperigillus flavus, Aspergillus candidus, and Penicillium cyclopium. Alternaria, Clado· sporium [22] and Aspergillus parasiticus [23] groups. While aflatoxin contamination in soybeans is not as prevalent as some other crops, such as peanuts, corn, and cottonseed (Department of Animal Science, Cornell, CALS. 2019), care must be taken to prevent damage to the seed coat and reduce elevated levels of humidity and moisture.

## 3. Soybean Processing

Globally, soybeans are grown predominantly for oil, with soybeans comprising approximately 90% of the U.S. oilseed production [24]. Soy oil processing plants are used to extract the oil directly from the bean by solvent extraction methods. Following the harvest, the collected soybeans are cleaned of all foreign material, such as sand, stems, sticks, leaves, and rocks (Figure 1). After cleaning, the beans are cracked under the pressure of corrugated rollers. This process creates bean particles of various sizes, and, subsequently, the beans are dehulled to remove the hull and outer husk (Figure 2). Approximately 7% of the soybean is the outer husk [25]. The removal of the outer husk increases the oil extraction efficiency. If not removed, the husk will retain oil in the pressed cake [25]. Additionally, dehulling reduces the fiber content while increasing the final protein content in the meal when separated following the oil extraction. Subsequently, dehulled soybeans are conditioned in rotating drums (Figure 2) using steam at a temperature setting of 149 °F [25]. It is during the conditioning and subsequent flaking process that the cells walls are ruptured to release the oil. This process involves the stretching and flattening of the soybean seed by increasing the surface area, which causes oil loss from the cells [25]. Lastly, oil is extracted from the conditioned flakes via successive hexane solvent washes. The extracted flakes are then desolventized to remove residual hexane, leaving the soybean cake (Figure 2), for use as the solvent extracted defatted soybean meal [26]. Hexane is removed from the extracted oil in rising film evaporators and with final vacuum distillation for use as conventional solvent extracted soybean oil [26].

Conventional solvent extracted soybean oil has a typical fatty acid composition of 51% linoleic acid (omega 6), 7–10% α-linolenic acid (omega 3), 23% oleic acid (omega 9), 10% palmitic, and 4% stearic saturated fatty acid composition [27]. However, current plant breeding programs have modified the fatty acid profile of new soybean cultivars

with reduced linoleic acid and increased oleic acid (C18:1 n-9) concentrations, producing high-oleic soybean cultivars (Table 1), with advantages as compared with conventional soybean cultivars, with increased oxidative stability, positive sensory taste profiles, and improved nutrition profiles for animal feed use [28]. Comparatively, conventional-oleic soybean cultivars possess higher levels of linoleic, palmitic, and stearic fatty acid levels and lower levels of oleic acid in comparison with high-oleic soybean cultivars (Table 1).

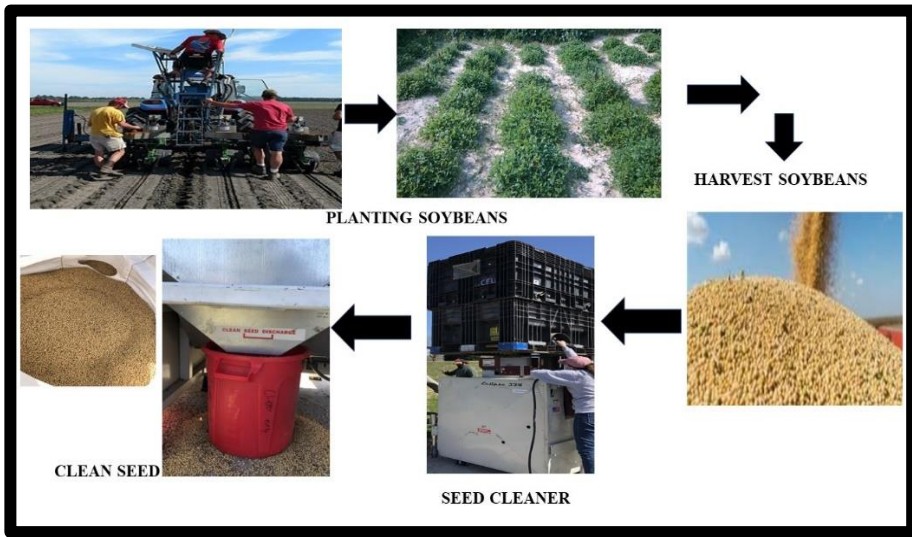

**Figure 1.** Soybean Cultivation and Harvest Processing. Soybean cultivars are cultivated and harvested. Post-harvest seeds are cleaned of all foreign material using an Eclipse 324 seed cleaner (images are taken from studies conducted by the Soybean Nitrogen and Fixation Unit, ARS, US Dept. of Agriculture, Raleigh, NC).

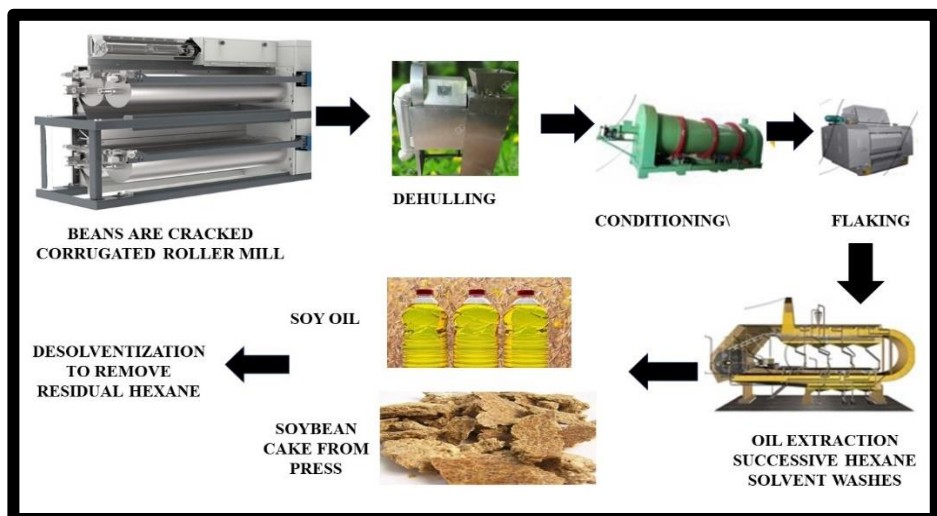

**Figure 2.** Post-Harvest Soybean Processing. Post-harvest raw soybeans are cracked using corrugated rollers, and are subsequently dehulled, conditioned, and flaked prior to oil extraction. Soy oil is extracted from the conditioned flakes via successive hexane solvent washes. Residual hexane is removed from soybean cake to produce edible solvent extracted defatted soybean meal and soy oil.

**Table 1.** Proximate Composition of Conventional and High-Oleic Soybeans.

| Parameters | Conventional-Oleic * | High-Oleic * |
|---|---|---|
| Crude protein (%) | 36.6 | 38.15 |
| Crude fat (%) | 17.94 | 16.38 |
| Gross energy (kcal/kg) | 5238 | 5236 |
| Palmitic acid (%) | 10.5 | 6.92 |
| Stearic acid (%) | 2.89 | 0.58 |
| Oleic acid (%) | 19.5 | 81.53 |
| Linoleic acid (%) | 51.56 | 4.76 |

* Conventional Oleic soybean varieties = NC-Roy-normal oleic, normal linolenic beans acquired from Soybean Nitrogen Fixation Unit, ARS, Raleigh, NC. High-Oleic soybean varieties = N16-1286 BC4 NIL-high oleic, low linolenic beans acquired from Soybean Nitrogen Fixation Unit, ARS, Raleigh, NC. The proximate composition analysis was performed using standard methods by an AOAC-certified lab using AOAC standard methods, ATC Scientific (Little Rock, AR, USA).

Hence, with the reduced competitive utilization of edible plant oils, such as rapeseed, soybean, and palm oil as biofuel [29], the use of full-fat whole soybeans in animal food production has gained new interest as a means to provide high quality, digestible protein and energy in the diets of poultry [30] and other livestock [31–33]. With dry extrusion processing of the whole soybean without oil extraction from the bean, a full-fat soybean meal (FFSBM) is produced, containing approximately 38–40% protein and 18–20% fat [34,35]. Quantitative analysis of various processed soybean meals demonstrate that the full-fat soybean meals provide the highest levels of gross energy and crude fat (Table 2), while extruded-expelled defatted soybean and solvent-extracted defatted soybean meal provide higher levels of dietary protein [36]. Interestingly, most recent nutrient digestibility studies conducted in layer hens fed diets containing either full-fat or defatted soybean meals [37] reported similar digestibility values of crude fat ranging from 71–84% and crude protein ranging from 67–72% ($p > 0.05$), with treatment mean values being different between the dietary treatments (Table 3). In parallel, the feeding trials conducted by [38] with Nile tilapia fed various feed ingredients to determine their digestibility characteristics, reported similar mean protein (87% full-fat soybean meal, 93% solvent extract soybean meal) and energy (75% full-fat soybean meal, 82% solvent extracted soybean meal) digestibility coefficient values for the full-fat soybean meal and solvent extract soybean meal.

Ravindran et al. [39] conducted a feeding trial with four samples of FFSBM taken from commercial feed mills in Southeast Asia for nutrient analysis, apparent metabolizable energy (AME), and ileal amino acids digestibility. As per these findings, FFSBM has greater AME than the defatted soybean meal (SBM), but less digestible contents of protein and amino acids. In the four samples, the crude protein, fat, AME, and standardized ileal digestibility coefficient of protein had a range of 351 to 399 gm/kg, 177 to 192 gm/kg, 12.62 to 15.46 MJ/kg and 0.763 to 0.821, respectively. While informative, these studies utilized a small sample size collected from commercial feed mills with an unknown processing history.

The soybean seeds contain approximately 35–40% carbohydrate content in the soybean meal, with half of this percentage present as non-structural oligosaccharides and a small percentage as pectic polysaccharides [40], with about 6% present as non-starch polysaccharide crude fiber [41] having anti-nutritive effects [42]. While soybeans are a nutritionally rich source of plant protein and essential amino acids, soybeans contain anti-nutritional factors, such as trypsin inhibitors (Kunitz trypsin inhibitors, Bowman-Birk trypsin inhibitors), lectins, saponins, and goitrogenic factors [43,44], which have been shown to reduce growth performance, reduce feed efficiency, and cause pancreatic enlargement and small-sized egg production in poultry [45].

## 4. Limitations/Anti-Nutritional Factors of Soybeans/Meal and Processing Methods

The raw soybean, if unprocessed, contains various anti-nutritional factors, such as trypsin inhibitors and ureases which significantly impact nutrient digestion and/or absorption, adversely affecting the animal growth performance (Table 4) and production [46,47]. Among the most important anti-nutritional factors (ANFs) are trypsin inhibitors and non-starch polysaccharides [46,48,49]. The two most crucial trypsin inhibitors, Bowman-Birk inhibitors and the Kunitz inhibitors, provide protection to the plant during germination from microorganisms within the soil and airborne pests before the maturity of the seed. Nevertheless, these protease inhibitors present in raw soybeans inhibit protein digestion/absorption and, thus, growth when consumed by animals unprocessed (Figure 3). Moreover, it has been demonstrated that the animals consuming unprocessed soybeans had pancreatic enlargement and overproduction of pancreatic trypsin and chymotrypsin [50]. Typically, the unprocessed soybean has a trypsin inhibitor (TI) activity of 20 to 35 mg/gm, while the recommended threshold level is 4 mg/gm in the diet, to prevent adverse growth performance [45]. Moreover, unprocessed soybeans containing TI negatively impacts nitrogen retention, with increased excretion of metabolic nitrogen [50].

**Table 2.** Comparative Chemical Composition of Various Processed Soybean Meals *.

| Parameters | Soybean Meal Variety | | | |
| --- | --- | --- | --- | --- |
| | **EECO** | **FFHO** | **FFCO** | **SECO** |
| Crude protein (%) | 43.83 | 39.86 | 39.56 | 45.74 |
| Crude fat (%) | 7.12 | 15.53 | 17.30 | 4.75 |
| Crude fiber (%) | 7.20 | 7.90 | 6.90 | 5.50 |
| Crude ash (%) | 6.02 | 5.10 | 5.39 | 6.16 |
| Moisture content (%) | 5.58 | 7.83 | 5.43 | 10.00 |
| Urease | 0.06 | 0.27 | 0.29 | 0.07 |
| Trypsin inhibitor (mg/g) | 7.64 | 6.92 | 7.99 | 2.40 |
| Gross energy (kcal/kg) | 4598 | 4890 | 4863 | 4105 |
| Palmitic acid (%) | 11.27 | 7.74 | 11.14 | 14.07 |
| Stearic acid (%) | 3.68 | 3.11 | 3.55 | 3.67 |
| Oleic acid (%) | 19.72 | 71.67 | 18.04 | 14.25 |
| Linoleic acid (%) | 53.21 | 11.03 | 55.20 | 55.87 |
| Formulated metabolizable energy (kcal/kg) | 2927 | 2927 | 2927 | 2927 |

EECO: extruded-expelled defatted soybean meal prepared from conventional oleic acid soybeans, FFHO: full-fat soybean meal prepared from high oleic soybeans. FFCO: full-fat soybean meal prepared from conventional oleic acid soybeans, SECO: solvent-extracted defatted conventional soybean meal prepared from conventional oleic soybeans. The proximate composition was analyzed using standard methodology by an AOAC-certified lab, ATC Scientific (Little Rock, AR, USA). * [36].

**Table 3.** Apparent Nutrient Digestibility of Full-fat and Defatted Soybean Meal in Layer Hens *.

| Parameters | TRT1 | TRT2 | TRT3 | TRT4 | SEM | *p*-Value |
| --- | --- | --- | --- | --- | --- | --- |
| DC, Fat | 0.841 | 0.842 | 0.749 | 0.711 | 0.067 | 0.44 |
| DC, Protein | 0.695 | 0.724 | 0.677 | 0.725 | 0.048 | 0.869 |
| AMEn (kcal/kg) | 2671 | 2764 | 2804 | 2745 | 51 | 0.356 |

* [37]. Digestibility coefficients (DC) for crude fat (i) and crude protein (CP) (ii) were not different ($p > 0.05$) between treatment diets. Nitrogen-corrected apparent metabolizable energy (AMEn) values for the diets were not different ($p > 0.05$). Four isonitrogenous and isocaloric treatment diets were fed to birds: Treatment (TRT) 1-Control = conventional diet containing solvent-extracted defatted soybean meal and corn; Treatment2 = diet containing extruded-expelled defatted conventional-oleic soybean meal and corn; Treatment3 = diet containing full-fat conventional-oleic soybean meal and corn; Treatment4 = diet containing full-fat high-oleic soybean meal and corn.

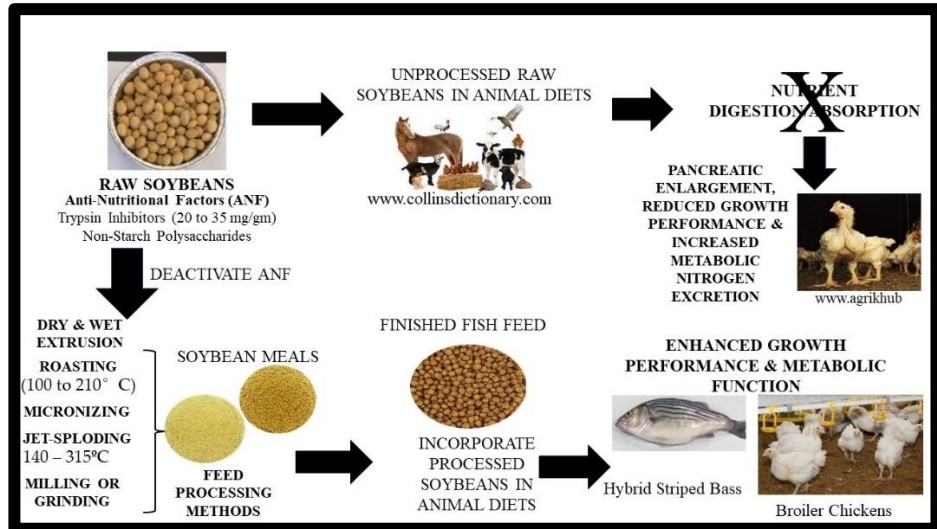

**Figure 3.** Soybean Post-harvest Processing and Utilization in Animal Diets. Soybean seeds were cultivated and harvested by the Soybean & Nitrogen Fixation Unit, ARS, U.S. Dept of Agriculture (Raleigh, NC). Raw soybeans were thermally processed in meals at Mule City Specialty Feeds (Benson, NC, USA) and finished fish feed was prepared at Zeigler Feeds (Gardners, PA). Domesticated juvenile striped bass were reared at the Marine Aquaculture Research Center at NC State University (Smyrna, NC). Male broiler (Ross 708) chickens were raised at the NC State University Chicken Education Research Center (Raleigh, NC).

Soybeans contain an ideal amino acid profile when fed in combination with corn in the diet, providing all the essential amino acids, apart from methionine. Methionine is a limiting amino acid in soybean, and this is resolved by the supplementation of synthetic methionine [30] to the diet. Nonetheless, even with dietary methionine supplementation, the finished feed diets containing raw or marginally processed soybeans compromise overall growth performance when fed to animals [51]. Additionally, raw/unprocessed soybeans contain low levels of the sulfur amino acids and moderate to high levels of trypsin inhibitors, which in combination greatly reduce the bioavailability of the amino acids in the soybean seed [52].

Trypsin inhibitors found in soybeans and soybean meal are dramatically reduced using thermal processing methods. Various thermal processing methods are routinely used in the preparation of commercial soybean meal, including dry and wet extrusion, roasting, and mechanical processing (milling, grinding) to deactivate trypsin inhibitors in animal diets (Figure 3). However, excessive utilization of heat and processing may greatly reduce the protein quality and bioavailability [53]. Hence, microbial feed enzymes (phytase and protease) are commonly utilized in finished feeds containing soybean meal to improve the nutrient digestibility by reducing the action of the non-heat labile anti-nutritional factors found in soybeans [53]. The ideal processing methods eliminate ANF, while not altering protein quality, amino acid content, or bioavailability of the proteins in the soybean meal.

Heat (thermal) treatment reduces and inactivates the ANFs in raw soybeans due to protein denaturation. Flaking, cooking, and roasting are common processing methods, which result in different nutrient profiles for the final products, due to differences in the processing temperatures. The simplest method of processing oilseeds is via "cooking", in which the raw soybeans are fully immersed in water and cooked from 30 min to 120 min, and, subsequently, mechanically or air-dried. Alternatively, raw soybeans and other oilseeds can be processed via roasting at temperatures ranging from 100 to 210 °C (Figure 3). The drying methods vary from heating systems using flame, coal burner, or oven for a minimum of approximately 20 s followed by milling. Flaking is a hydro-thermal processing method which involves the injection of low-pressure steam into a conditioner to cook the

bean. During this process, oil is readily released from pressing the beans between two rollers, which transforms the soybeans into flakes [54].

Extrusion, micronizing, and jet-sploding are additional processing methods used frequently with soybeans and other oilseeds. The extrusion process involves a high temperature (80–200 °C) and short time interval of 10–270 s [55]. In the context of soybean extrusion using a high shear process, Albin [56] states the residence time of 15 s with the maximum temperature encountered for about 5 s. The extrusion process can be divided into two types—dry processing and wet processing. Dry processing uses the frictional heat generated between soybeans and the extruder barrel as it travels forward within the barrel. Wet processing uses steam in addition to the mechanical energy. The steam can be added into the preconditioner (if available) or can be directly added to the extruder barrel itself. The processing temperatures are different for both methods, with wet extrusion running at lower temperature of 135–140 °C [56]. This is due to it having higher moisture content and, thus, lower mechanical or frictional energy as compared to dry extrusion, which involves temperatures of 150–160 °C [57,58]. During dry extrusion, the pressure inside the extruder barrel is very high; around 40 atm [57]. Dry extrusion will generally have higher screw speeds to generate the additional shear needed to properly process soybeans, as compared to wet extrusion. However, wet extrusion, i.e., when the raw material is preconditioned with steam, can almost double the efficiency of the extruder and reduce the wear of extruder barrel components by 20% [58], due to the decrease in frictional resistance. The impact on the quality of the soybean meal produced between these two methods is due to the presence of moisture during the wet processing, which interferes with the mechanical oil extraction. Furthermore, full-fat soybean meal produced using dry extrusion has a final moisture content between 5–6%, compared with around 10% using the wet extrusion method [59]. Micronizing is an infrared dry heating method in which radiating heat is used as an energy source to thermally process grains [54,57]. Micronizing involves the use of heating ceramic plates using electronic or gas burners (Figure 3). The heated ceramic plates emit infrared dry radiant heat onto the beans. The micronizing process is highly efficient in thermally penetrating the beans, increasing the bean's internal temperature. Jet-sploding is another alternative processing method for oilseeds, such as soybeans, to inactivate trypsin inhibitors (Figure 3). Jet-sploding involves exposing the raw beans to a stream of pre-heated air at temperatures between 140–315 °C. This process causes the grain to heat from the inside-out, attaining a core temperature of 90–95 °C, which causes the intracellular water to boil and cook the grain. After heating, the beans are mostly shifted into a cylinder mill to complete the process and ease the release of the intercellular fat [54].

## 5. Soybean Meal Anti-Nutritional Quality Control Methods

Anti-nutritional factors present within thermally unprocessed soybean meal is greatly reduced with mechanical heat or thermal heat processing. Nevertheless, tight quality control measures must be adhered to, to prevent damage and loss of the bioavailability of amino acids due to the occurrence of Maillard reaction occurring between the aldehyde group of sugar moieties and free amino groups [60] with excessive heat or mechanical processing, while providing adequate processing to inactivate anti-nutritional factors within the meal. Therefore, several official analytical methods are commonly utilized to test the quality of processed soybean meal, which includes the nitrogen solubility index (NSI), protein dispersibility index (PDI), and urease activity (UA) (Table 4). Unofficial analytical methods commonly used to test the quality of processed soybean meal include digestible/reactive lysine, protein solubility in potassium hydroxide, and trypsin inhibitor activity [60].

Urease content is variable in raw soybeans and is not of a nutritional significance. Nevertheless, the soybean urease content can be utilized as an indirect marker to assess soybean thermal processing (Table 5). Urease is a heat-labile enzyme present in raw soybeans responsible for the conversion of urea to ammonia and is destroyed during thermal processing and can be correlated to the destruction of trypsin inhibitors and lectins.

Urease activity is a widespread method used worldwide as an indicator of the quality of the processed full-fat soybean meal (FFSBM) due to its simplicity. The AOCS Official Method Ba. 9–58 which is used for UA determination is based on the measurement of change in pH units.

While UA is a very common analysis used to test the quality of soybean, this assay is best utilized to examine the quality of under-processed soybean meal and may not be efficient to accurately determine the quality of over-processed soybean meal. The protein solubility (PS) in the KOH assay is used to test the quality of over thermally or mechanically processed soybean meal, with very high values being indicative of under-processed meal (Table 6). Generally, the KOH solubility declines as the points of the heat treatment surges [61]. Nonetheless, these analysis methods when combined with other tests including the protein dispersibility index (PDI) and nitrogen solubility index (NSI), are beneficial to determine the quality of the soybean or soybean meal.

**Table 4.** Urease activity of FFSBM Processed at Different Temperatures and Laboratories on Body Weight Gain (BWG) and Feed Conversion Ratio (FCR) of Chickens from 0–14 days of age *.

| Heat (°C) | BWG (g) | FCR (kg/kg) | UA (Δ pH) | |
| --- | --- | --- | --- | --- |
| | | | [1] Lab 1 | [1] Lab 2 |
| 115 | 92.2 [bc] | 1.953 [bc] | 2.189 [a] | 1.876 [b] |
| 125 | 105.1 [b] | 1.735 [c] | 0.433 [a] | 0.239 [c] |
| 135 | 135.5 [a] | 1.350 [a] | 0.080 [c] | 0.069 [c] |
| 145 | 138.6 [a] | 1.335 [a] | 0.026 [c] | 0.044 [c] |
| 165 | 85.3 [c] | 1.899 [c] | 0.028 [c] | 0.035 [c] |

* [60]. [abc] Means without common superscript in the same row differ significantly ($p < 0.01$). UA = urease activity in change of pH units in FFSBM (full-fat soybean meal). BWG = body weight gains. FCR = feed conversion rate (total feed consumed/total carcass weight). [1] Lab 1 and Lab 2 = means of values from 7 replicates obtained by two analysts in different laboratories. Data analyzed by Student's $t$ test.

**Table 5.** Relation between the Temperature of Extrusion, Degree of FFSBM Processing and Urease *.

| Temperature of Extrusion (°C) | Degree of FFSB Processing | UA (Δ pH) |
| --- | --- | --- |
| <135 | Under–processed | >0.20 |
| 135–145 | Adequately processed | 0.05–0.20 |
| >145 | Over–processed | <0.05 |

* [60]. Globally accepted relation between the degree of full-fat soybean meal (FFSB) processing and urease activity (UA) expressed as change in pH units.

The experiments conducted by [62] in which soybean was extruded at different temperatures and subsequently expelled revealed that the meal produced at the highest temperature of 160 °C had a lower than desirable PS and PDI; however, the amino acid digestibility was elevated at that temperature. The meal produced at a temperature of 121 °C and 135 °C was under-processed due to the high urease value (Table 5) and less amino acid digestibility. The protein solubility was average but the phytate phosphorus was lowest at 160 °C. Overall, it can be concluded that extruded and expelled soybean meal have somewhat different quality parameters as compared with solvent-extracted soybean meal or extruded full-fat soybean meal. The amino acid digestibility is the best indicator for the quality of the soybean meal [63]. Ruiz's [64] findings also give emphasis that a KOHPS test is not a trustworthy sign of lysine digestibility for the full-fat soybean meal.

The Cresol Red Test is a rapid, semi-quantitative method to measure the protein of thermally-processed soybean meal. In this colorimetric assay, soybean meal proteins absorb cresol red dye, with increasing dye absorption with increasing thermal processing (Table 7), which can be utilized as an indirect indicator for the protein content and quality. In general, cresol red levels below 3.7 mg/gm are indicative of under-processed soybean meal, while

cresol red levels of 3.7 to 4.3 mg/gm are indicative of adequately processed soybean meal, and 4.3 to 4.5 mg/gm cresol levels of over-processed soybean meal [65]. Nevertheless, the cresol test does not measure anti-nutritional factors present in the processed soybean meal.

**Table 6.** Quality indices of raw soybeans and extruded soybeans at different die temperatures *.

| Indices | Target Range | Raw Soybeans | Extruded Soybean Meal Processing Temperatures | | | | | |
|---|---|---|---|---|---|---|---|---|
| | | | 135 °C | 145 °C | 155 °C | 160 °C | 165 °C | 170 °C |
| Dry matter (%) | - - - | 90.19 | 94.68 | 95.07 | 95.60 | 95.75 | 96.15 | 96.43 |
| Crude protein (%) | - - - | 37.59 | 40.57 | 41.74 | 41.59 | 41.59 | 43.85 | 45.23 |
| Lysine (%) | - - - | 2.45 | 2.65 | 2.54 | 2.61 | 2.61 | 2.65 | 2.71 |
| PDI (%) | 30–35 | - - - | 40.27 | 36.05 | 33.47 | 33.47 | 28.61 | 26.47 |
| KOH protein solubility (%) | <73 | 77.07 | 79.09 | 73.50 | 74.57 | 74.57 | 68.29 | 57.04 |
| Urease index (U) | 0.05–0.3 | 2.09 | 0.08 | 0.04 | 0.02 | 0.02 | 0.03 | 0.04 |
| Trypsin inhibitor | 1–3.5 | 2.44 | 3.76 | 3.91 | 3.65 | 3.52 | 2.26 | 0.04 |
| Lys:CP ratio | >6 | 6.50 | 6.53 | 6.07 | 6.26 | 6.43 | 6.04 | 5.99 |

PDI = Protein Dispersibility Index (measurement of soybean meal protein dispersed in water after blending a sample with water); KOH = Potassium Hydroxide protein solubility (indices of protein nutritional quality); Urease index (measurement of urease enzymatic activity, urease an enzyme that catalyzes and denatures proteins); Trypsin inhibitor (protease inhibitors naturally found in soybeans that reduces the biological activity of trypsin and prevent the digestion and absorption of dietary proteins); U = Urease index units; Lys = Lysine. CP = Crude protein; Lys:CP ratio used as a measure of nutritive value of soybean meal protein. * [62].

**Table 7.** Poultry Apparent Metabolizable Energy (AME), Nitrogen Retention (NR), and Cresol red absorption of Processed FFSBM *.

| Process of FFSM | AME (Kcal/kg) [1] | NR (%) [1] | Cresol Red Absorption (%) [2] |
|---|---|---|---|
| Wet Extrusion | 4278 | 54 | 4.60 |
| Dry Extrusion | 4159 | 59 | 4.06 |
| Micronized | 3681 | 48 | 4.00 |
| Jet-Sploded | 3513 | 61 | 3.98 |
| Toasted | 3728 | 57 | 3.81 |
| Raw | 3227 | 30 | 2.50 |

* [34] [1], [66] [2]. Experiments were conducted at the University of Nottingham concerning the influence of processing on the energy value of full-fat soybean meal (FFSBM) in 18-day-old chicks. NR = nitrogen retention (dietary nitrogen intake-nitrogen content ileal content), AME = Apparent Metabolizable Energy corrected by nitrogen calculated using the following formula: GE (feed) − [GE (fecal) × (acid insoluble ash recovery feed)/acid insoluble ash recovery fecal) − (8.22 × (crude protein fecal/6.25). Cresol red absorption was conducted to quantitative protein of various thermally processed soybean meal.

## 6. Conclusions

Today, soybean farmers use innovative farming strategies and technology and plant new varieties that yield healthier crops, while embracing technology and environmentally sustainable practices to protect the land for future generations, while meeting the growing demand for high-quality plant proteins. Moreover, the benefits of soybeans have also been maximized with optimized processing methods using quality control parameters (urease content, content trypsin inhibitors, KOH protein solubility, protein dispersibility index, nitrogen solubility index, Cresol red absorption-protein) to produce defatted meal and soy oil. Even so, much research is left to be determined with effective processing methods and quality control measures for the optimal meal preparation. Futuristically, the soybean production will continue to expand, with increasing global demands for the use of

soybeans in industrial applications, as biofuel, food to meet the needs of a growing global population, animal feed in animal food production, driving soybean research, innovation farming methods, and the use of higher yielding resistant seed varieties.

**Funding:** This research was funded by United Soybean Board grant number 1930-362-0618 and the Agricultural Research Service, US Department of Agriculture by appropriated funds CRIS 6070-43440-013-00D.

**Acknowledgments:** The authors would like to acknowledge Philip Lobo and the United Soybean Board-Animal Nutrition Working group for providing their guidance and leadership, and Muquarrab Qureshi and the SEA ARS Location for administrative support and leadership.

**Conflicts of Interest:** The authors declare no conflict of interest.

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
