# Peer review of "Current Agronomic Practices, Harvest & Post-Harvest Processing of Soybeans (Glycine max)—A Review†"

_agronomy, doi:10.3390/agronomy13020427_

Round 1

Reviewer 1 Report

- MDPI format is not respected.

- design a graphical abstract, this is really useful for a review article.

- Keywords should not be included in the title and should not be repeated. write them again

- Articles of 30 years ago? My main question to the authors is after reviewing this manuscript what did you achieve?  A good review article should include the latest developments in the field. at least 50% of your references should be after 2018. What are you looking for in reviewing the articles of the last 30 years?

- Line 118: Heterodera glycines should change to H. glycines

- Line 126: Soybean Cyst Nematode should change to SCN because you mentioned it in parentheses before. Check all abbreviations once again.

- Also, you can show the soybean processing with a figure in section 3

- Draw a figure for section 4. Let's make the manuscript interesting by designing what you are saying.

- Tables 6 & 7: can you find updated data for these two tables?

- Using reference in conclusion!? I have never seen it before. I think you should write this part by summarizing the contents and your knowledge.

Author Response

The authors' would like to thank the reviewer for their constructive comments. 

Please find below the author's rebuttal to each of reviewer 1 comments.

Reviewer 1 Author Response to Comments Soybean Review1 Manuscript

Reviewer 1 Comments

- MDPI format is not respected.

Author response: Manuscript was edited to comply with MDPI format.

- design a graphical abstract, this is really useful for a review article.

Author response: Per the reviewers comments a graphical abstract has been included in the edited submission.

- Keywords should not be included in the title and should not be repeated. write them again

Author response: Per the reviewers comments the keywords in the edited manuscript have been edited to the following: precision farming techniques, Soybean meal quality; Extrusion processing; high-oleic soybeans

- Articles of 30 years ago? My main question to the authors is after reviewing this manuscript what did you achieve?  A good review article should include the latest developments in the field. at least 50% of your references should be after 2018. What are you looking for in reviewing the articles of the last 30 years?

Author response: Per the reviewers comments the references listed below from the 70’s, 80’s, 90’s have been replaced with updated references. Please see below:

Ogundipe, S.O.; Adams, A.W. Practical, raw-soybean diets for egg-type pullets. Poult. Sci. 1974, 53, 2095-2101.

Replace with Bernard, J.K. Oilseed and Oilseed Meals. 2022 Elsevier Ltd. The University of Georgia, Department of Animal and Dairy Science, Tifton, GA, United States

23 Mislivec, P.B.; Bruce, V.R. Direct Plating versus dilution plating in qualitatively determining the mold flora of dried beans and soybeans. J. Assoc. Off. Anal. Chem. 1977, 60, 741-743.

Mannaa, M.; Kim, K.D. Effect of Temperature and Relative Humidity on Growth of Aspergillus and Penicillium spp. and Biocontrol Activity of Pseudomonas protegens AS15 against Aflatoxigenic Aspergillus flavus in Stored Rice Grains. Mycobiology. 2018, 46, 287–295, doi: 10.1080/12298093.2018.1505247

  1. Serrato, A.G. Extraction of oil from soybeans. J. Am. Oil Chem. Soc. 1981, 58, 157-159, https://doi.org/10.1007/BF02582327.

Replace with Food Processing Technologies. 2022. Soybean Processing. Overview. http://foodtechinfo.com/foodpro/facility_types/311222_soybean_processing/. Accessed online January 14, 2023

  1. Liener, I.E. Factors affecting the nutritional quality of soya products. J. Am. Oil Chem. Soc. 1981, 58, 406–413.

Replace with Bueno, R.D.; Borges, L.L.; Good God, P.I.V.; Piovesan, N.D.; Teixeira, A.L.; Damião Cruz, C.; DE Barros, E.G. Quantification of anti-nutritional factors and their correlations with protein and oil in soybeans. An Acad Bras Cienc. 2018, 90, 205-217. doi: 10.1590/0001-3765201820140465

  1. Waldroup, P.W. Whole soybeans for poultry feeds. Worlds Poult. Sci. J. 1982, 37, 28–35.

Replace with Yang, H.W.; Hsu, C.K.; Yang, Y.F. Effect of thermal treatments on anti-nutritional factors and antioxidant capabilities in yellow soybeans and green-cotyledon small black soybeans. J Sci Food Agric. 2014, 94, 1794-1801. doi: 10.1002/jsfa.6494

  1. Grant, G. Anti-nutritional effects of soya bean: a review. Prog. Food Nutr. Sci. 1989, 13, 317–348.

Replace with Aderibigbe, A.S.; Cowieson, A.J.; Ajuwon, K.M.; Adeola, O. Contribution of purified soybean trypsin inhibitor and exogenous protease to endogenous amino acid losses and mineral digestibility. Poult Sci. 2021, 100, 101486.  https://doi.org/10.1016/j.psj.2021.101486

Hence in 1996, Monsanto genetically engineered “Roundup Ready Soybeans” which are resistant to glyphosate [14].

  1. Parry, Mark E. Monsanto—The Launch of Roundup Ready Soybeans. Darden Case No. UVA-M-0619, Available online: http://dx.doi.org/10.2139/ssrn.910079 (accessed on 30 November 2022)

Author response: No replacement for this reference is provided in the edited manuscript because this is a historical statement made in the review.

Varco JJ. Nutrition and fertility requirements. In Soybean Production in the Mid-South, 1st ed.; CRC Press: Boca Raton, FL, USA, 1999; pp. 53-70.

Replace with Flynn, R.; Idowu, J. 2015. Nitrogen Fixation by Legumes. New Mexico State University Cooperative Extension Service. College of Agricultural, Consumer and Environmental Sciences. Guide A-129. https://pubs.nmsu.edu/_a/A129/ Accessed online January 14, 2023

Preissinger, W.; Schwarz, F.J.; Kirchgessner, M. Feed intake and milk production in dairy cows fed whole fat soybeans. Arch Tierernahr. 1997, 50, 347-359, doi: 10.1080/17450399709386144.

Replace with Zanferari,F.; Vendramini, T.H.A.; Rentas, M.F.; Gardinal, R.; Calomeni, G.D.; Mesquita, L.G.; Takiya, C.S.; Rennó, F.P. Effects of chitosan and whole raw soybeans on ruminal fermentation and bacterial populations, and milk fatty acid profile in dairy cows. J Dairy Sci. 2018, 101, 10939-10952, doi: 10.3168/jds.2018-14675

Liener, I.E. Implications of antinutritional components in soybean foods. J. Crit. Rev. Food Sci. Nutr. 1994, 34, 31–67

Replace with Takács, K.; Szabó, E.E.; Nagy, A.; Cserhalmi, Z.; Falusi, J.; Gelencsér, E. The effect of radiofrequency heat treatment on trypsin inhibitor activity and in vitro digestibility of soybean varieties ( Glycine max. (L.) Merr.). J Food Sci Technol. 2022, 59, 4436-4445, doi: 10.1007/s13197-022-05523-z

- Line 118: Heterodera glycines should change to H. glycines

Author response: Edits made per the reviewers’ comments on line 112 and 113 of the edited manuscript.

- Line 126: Soybean Cyst Nematode should change to SCN because you mentioned it in parentheses before. Check all abbreviations once again.

Author response: Line 122 “the reproduction of the soybean cyst nematode and improves yield in soybean plants “edited to “the reproduction of the SCN and improves yield in soybean plants”

- Also, you can show the soybean processing with a figure in section 3

Author response: Per the reviewer’s comments Figure 1 and Figure 2 have been incorporated into Section 3 of the review. Images in Figure 1 were taken from soybean processing and feeding studies conducted within the Food Science & Market Quality and Handling Research Unit and the Soybean Nitrogen Fixation Unit in Raleigh, North Carolina. Images in Figure 2 were taken from the public domain.

- Draw a figure for section 4. Let's make the manuscript interesting by designing what you are saying.

Author response: Per the reviewer’s comments Figure 3 has been incorporated into Section 4 of the review. Images from Figure 3 were taken from soybean processing and feeding studies conducted within the Food Science & Market Quality and Handling Research Unit and the Soybean Nitrogen Fixation Unit in Raleigh, North Carolina.

- Tables 6 & 7: can you find updated data for these two tables?

Author response: Table 6 has been replaced with data from a more recent publication published in 2022 in the edited manuscript. See cited reference below.

  1. Patino, D.; Joseph, M. Ideal extruder temperature to produce best full fat soybean meal. Feed and Additive Magazine, International Magazine for Animal Feed & Additives Industry. 2022, 70-73, Available online. https://www.feedandadditive.com/ideal-extruder-temperature-to-produce-best-full-fat-soybean-meal/ (accessed on 19 January 2023).

Author response: In table 7 of the originally submitted manuscript, one of the references was inadvertently omitted, which is a 2006 published reference containing the Cresol red absorption data which corresponds to the process of FFSM found in Table 7. This has been updated in the table as seen below and appears as reference number 68 in the reference list of the edited manuscript.

Table 7

Table 7. Poultry Apparent Metabolizable Energy (AME), Nitrogen Retention (NR) and Cresol red absorption of Processed FFSBM*.

Process of FFSM

AME (Kcal / kg)1

NR (%)1

Cresol red absorption (%)2

Wet Extrusion

4,278

54

4.60

Dry Extrusion

4,159

59

4.06

Micronized

3,681

48

4.00

Jet-Sploded

3,513

61

3.98

Toasted

3,728

57

3.81

Raw

3,227

30

2.50

 *[34]1, [68]2.

- Using reference in conclusion!? I have never seen it before. I think you should write this part by summarizing the contents and your knowledge.

Author response: In the conclusion (pasted in blue font below), the authors aimed to conclude with discussion of the past and the future regarding soybean research and soybean production. Therefore, the authors cited references regarding “How has demand for soy changed over time?” by Ritchie and Roser and from the United Soybean Board on “Forging the Future” (reference citations below in blue font). These references were used to support the statistics stated in the conclusion of the historical facts stated.

However, in response to the reviewers’ comment. These references and statements have been removed from the conclusion.

Global soybean production has grown by more than 13 times since the early 1960s [66]. In 2019, U.S. farmers grew over 3 billion bushels (81.5 million tons) of soybeans producing a third of the global market of soybeans, with 97% of the meal produced utilized for animal nutrition [67]. T

  1. Ritchie, H.; Roser, M.; Soy- How has demand for soy changed over time? Our World in Data. Available online: https://ourworldindata.org/soy (accessed on 29 November 2022).
  2. United Soybean Board, 2022. Forging the Future of U.S. Soy. https://www.unitedsoybean.org/hopper/soys-long-term-outlook/.Accessed online November 29, 2022.

Reviewer 2 Report

I read this Manuscript with interest. Overall, the information is well presented, and the Manuscript is easy to read. The Manuscript is a literature revision and presents technical and scientific merits and interesting information about the current agronomic practices, harvest and post-harvest processing of soybeans (Glycine max). Thus, I think the Manuscript falls within the scope of the Agronomy.

Title is adequate. Abstract and Introduction are good. Results and Discussion is well focused and combined with the literature. Conclusion is pertinent and in accordance with the objectives of the Manuscript. References are current and closely related to the scope of the Manuscript.

Author Response

Reviewer 2 Author Response to Comments Soybean Review1 Manuscript

Reviewer2 Comments

I read this Manuscript with interest. Overall, the information is well presented, and the Manuscript is easy to read. The Manuscript is a literature revision and presents technical and scientific merits and interesting information about the current agronomic practices, harvest and post-harvest processing of soybeans (Glycine max). Thus, I think the Manuscript falls within the scope of the Agronomy.

Title is adequate. Abstract and Introduction are good. Results and Discussion is well focused and combined with the literature. Conclusion is pertinent and in accordance with the objectives of the Manuscript. References are current and closely related to the scope of the Manuscript.

Author response: The authors thank the reviewer for the positive comments of the review. We greatly appreciate your review of the manuscript and your support.

Reviewer 3 Report

The review discusses an important topic. I recommend the publishing of this article; however, I have some minor comments.

Please explain more detailed the specific new aspects of your review and formulate a clear aim. This is also in close connection to my major concern. I miss, unfortunately, a methodological chapter.

Please consider important methodological approaches for systematic reviews as given in Liberati et al., 2009, Moher et al., 2009 or Shea et al., 2007.

You must ensure that another team of authors is able to reconstruct your search, your selection of studies and the outcomes you searched for in the studies (not least to do a comparable work in ten years, looking what happened since your review was completed):

 (a) In this chapter you have to clarify how you gained the considered publications. Who did the search? One or more persons? In which period did you conduct the search? With which search terms and on what platforms?

(b) How did you proceed the publications? Please define at least some outcome parameters you systematically searched for in the publications.

Author Response

Reviewer 3 Author Response to Comments Soybean Review1 Manuscript

Reviewer3 Comments

The review discusses an important topic. I recommend the publishing of this article; however, I have some minor comments.

Please explain more detailed the specific new aspects of your review and formulate a clear aim. This is also in close connection to my major concern. I miss, unfortunately, a methodological chapter.

Author response: This review is a literature review and not an empirical review of the existing literature data. As a consequence, there is not a “materials and methods” section.

Please consider important methodological approaches for systematic reviews as given in Liberati et al., 2009, Moher et al., 2009 or Shea et al., 2007.

Author response: This review is only a literature review and not a systematic review because the authors want to provide the readers with a general understanding or overview of the topic. To enable this, the authors have tried to summarize the literature under various topics within the review article. While the reviewer's suggestion is good, the authors are of the considered opinion that this topic of review would be better served with a general literature review."

You must ensure that another team of authors is able to reconstruct your search, your selection of studies and the outcomes you searched for in the studies (not least to do a comparable work in ten years, looking what happened since your review was completed):

Author response:

 (a) In this chapter you have to clarify how you gained the considered publications. Who did the search? One or more persons? In which period did you conduct the search? With which search terms and on what platforms?

Author response: This review was written as a historical literature review to cover the progress of soybean research and production over the past 20 years to current practices. As a consequence, the references cited span a time from 2 publications cited in the 1990’s (Wiseman, J., 1994. And Parry, Mark E. 1996 with Monsanto) and 9 cited publications between 2004 to 2009, with the remaining reference citations within the last ten years. This review also aimed to include current practices of soybean production, processing and research that have not been included in previous published soybean review articles.

Searches within these databases were conducted by Ondulla Toomer, Rouf Mian, Michael Joseph, Danny Patino, and Ali Muhammad. Soybean cultivation and feeding trials that are presented in the figures and/or data tables were conducted in studies by Ali Muhammad, Danny Patino, Edgar Oviedo, Ondulla Toomer, Rouf Mian, Mike Frinsko, Thien Vu, Pramir Maharjan, and Ben Fallen.

(b) How did you proceed the publications? Please define at least some outcome parameters you systematically searched for in the publications.

Author response: Literature searches were conducted PubMed and Science Direct databases using several key word searches such as: soybean production, crop production, soybean processing, soybean antinutritional factors, extrusion technology, oilseeds and oilseed meals, soybean production and germplasm, quality control soybean meal, full-fat soybean meal, extrusion processing, precision farming technologies, whole soybeans in the diets of animals, soybean meals and animal nutrition, soybean farm production and harvest, soybean agronomy

Please explain more detailed the specific new aspects of your review and formulate a clear aim. This is also in close connection to my major concern. I miss, unfortunately, a methodological chapter.

Author response: This review is a literature review and not an empirical review of the existing literature data. As a consequence, there is not a “materials and methods” section. The aspects of the review that provide “new” aspects of soybean research are the following: 1.) the use of full-fat soybean meal in the animal diets and 2.) the quality control measurements used to access the thermal processing of full-fat soybean meal to optimize processing methods to improve the nutritional quality of full fat soybean meal for use in animal diets, 3) the introduction of high-oleic soybean cultivars and use of full-fat high oleic soybean meal in animal diets, 4) the various tools of precision farming in soybean production not discussed in earlier published soybean reviews.

Round 2

Reviewer 1 Report

The current version looks acceptable.